# Tixagevimab/Cilgavimab as SARS-CoV-2 Pre-Exposure Prophylaxis in Lung Transplant Recipients during the Omicron Wave: A Real-World Monocentric Experience

**DOI:** 10.3390/microorganisms12071436

**Published:** 2024-07-15

**Authors:** Andrea Cona, Alessandro Tavelli, Stefano Agrenzano, Neha Hafeez, Giovanni Scianna, Angelo Maria, Francesco Marino, Elizabeth De La Cruz, Maria Di Giorgio, Eglys Osorio, Giuseppe Cucinella, Angelo Luca, Alessio Provenzani, Patrizio Vitulo, Alessandro Bertani, Paolo Antonio Grossi, Alessandra Mularoni

**Affiliations:** 1Department of Infectious Diseases, Mediterranean Institute for Transplantation and Advanced Specialized Therapies (IRCCS ISMETT), Via Ernesto Tricomi 5, 90127 Palermo, Italy; stefano.agrenzano@gmail.com (S.A.); amularoni@ismett.edu (A.M.); 2Unit of Infectious Diseases, ASST Santi Paolo e Carlo, Department of Health Sciences, University of Milan, 20146 Milan, Italy; alessandro.tavelli@gmail.com; 3Infectious Diseases Unit, ARNAS “Civico-Di Cristina-Benfratelli”, 90127 Palermo, Italy; 4Department of Medicine, University of Pittsburgh School of Medicine and University of Pittsburgh Medical Center, Pittsburgh, PA 15213, USA; hafeezn@upmc.edu; 5Transplant Coordinator Nurses, Mediterranean Institute for Transplantation and Advanced Specialized Therapies (IRCCS ISMETT), Via Ernesto Tricomi 5, 90127 Palermo, Italy; gscianna@ismett.edu (G.S.); amaria@ismett.edu (A.M.); 6Nursing Services, Mediterranean Institute for Transplantation and Advanced Specialized Therapies (IRCCS ISMETT), Via Ernesto Tricomi 5, 90127 Palermo, Italy; fmarino@ismett.edu (F.M.); edelacruz@ismett.edu (E.D.L.C.); mdigiorgio@ismett.edu (M.D.G.); eosorio@ismett.edu (E.O.); gcucinella@ismett.edu (G.C.); 7Department of Diagnostic and Therapeutic Services, Mediterranean Institute for Transplantation and Advanced Specialized Therapies (IRCCS ISMETT), Via Ernesto Tricomi 5, 90127 Palermo, Italy; aluca@ismett.edu; 8Clinical Pharmacy, Mediterranean Institute for Transplantation and Advanced Specialized Therapies (IRCCS ISMETT), Via Ernesto Tricomi 5, 90127 Palermo, Italy; aprovenzani@ismett.edu; 9Division of Pulmonology, Chest Center Department, Mediterranean Institute for Transplantation and Advanced Specialized Therapies (IRCCS ISMETT), Via Ernesto Tricomi 5, 90127 Palermo, Italy; pvitulo@ismett.edu; 10Division of Thoracic Surgery and Lung Transplantation, Chest Center Department, Mediterranean Institute for Transplantation and Advanced Specialized Therapies (IRCCS ISMETT), Via Ernesto Tricomi 5, 90127 Palermo, Italy; abertani@ismett.edu; 11Infectious and Tropical Diseases Unit, Department of Medicine and Surgery, University of Insubria-ASST-Sette Laghi, 21100 Varese, Italy; paolo.grossi@uninsubria.it

**Keywords:** COVID-19, SARS-CoV-2 pre-exposure prophylaxis, tixagevimab/cilgavimab, lung transplantation

## Abstract

Lung transplant recipients (LTRs) respond poorly to vaccination. SARS-CoV-2 pre-exposure prophylaxis (PrEP) with tixagevimab/cilgavimab (TIX/CIL) reduces the incidence of infection and the evolution to severe COVID-19. In vitro data show decreased activity against Omicron variants. We evaluated the clinical efficacy and safety of TIX/CIL in LTRs during the Omicron wave. A prospective observational cohort study was conducted at ISMETT in Palermo (Italy). In June 2022, SARS-CoV-2 PrEP with TIX/CIL 150/150 mg was offered to LTRs. LTRs who received TIX/CIL were compared to LTRs who did not. Logistic regression analysis (adjusted for prior COVID-19, SARS-CoV-2 vaccination, age, years from transplant, and rejection) was performed. The objective of this study was to compare the following among the two populations: prevalence of SARS-CoV-2, length of SARS-CoV-2 positivity, and COVID-19 disease severity. Among 110 eligible LTRs, 79 (72%) received TIX/CIL and 31 (28%) did not. SARS-CoV-2 infections occurred in 6% (n = 5) of patients who received TIX/CIL and 29% (n = 9) of patients who did not (*p* < 0.001). In both groups, infections were mild/asymptomatic, and no one was hospitalized or died. At multivariate analysis, TIX/CIL was associated with a lower risk of infection (aOR 0.22; 95%CI 0.06–0.78; *p* = 0.02). TIX/CIL was safe and effective in reducing the risk of SARS-CoV-2 in LTRs during the Omicron wave.

## 1. Introduction

Lung transplant recipients (LTRs) receive lifelong immunosuppressive therapy to prevent organ rejection but, as a consequence, are more prone to infections including the severe acute respiratory syndrome coronavirus 2 (SARS-CoV-2). Therefore, adequate prevention strategies, such as vaccination, are crucial for mitigating the post-transplant infectious risk. The clinical spectrum of COVID-19 ranges from asymptomatic infection to severe disease. As compared to other solid organ transplant (SOT) recipients, LTRs have an increased risk of COVID-19 following SARS-CoV-2, due to their higher level of immunosuppression [1] and because the lungs are the primary organs affected by SARS-CoV-2. COVID-19 vaccination is crucial for preventing infection and reducing progression towards severe forms. However, immunocompromised hosts respond poorly to vaccination and might fail to mount an adequate humoral and cellular immune response to vaccines, especially in the early period post-transplant when immunosuppression is higher [2,3,4]. Moreover, some patients may be unvaccinated or just partially vaccinated. Therefore, alternative strategies of prevention were developed including mask wearing and social distancing, booster vaccine doses, temporary reduction in the immunosuppressive regimen, and passive immunization with human monoclonal antibodies (mAbs).

Pre-exposure prophylaxis (PrEP) with tixagevimab/cilgavimab (TIX/CIL), a combination of two neutralizing human mAbs, has been proven to reduce the incidence of breakthrough infection, evolution to severe COVID-19, and risk of hospitalization [5,6,7,8], especially in immunocompromised patients [9,10,11,12,13,14]. The efficacy and safety of TIX/CIL, also known as AZD7442, has been investigated in several studies. In the PROVENT study, a Phase III clinical trial which included 253 kidney transplant recipients among more than 5000 participants, administration of TIX/CIL reduced the risk of SARS-CoV-2 infection by 77% with no cases of severe COVID-19 or COVID-19-related deaths [5]. These results have been confirmed by real world data in SOT recipients [15,16]. The efficacy in observational studies has also been proven for other categories of immunocompromised hosts, such as patients with solid or hematological malignancies, and those on B-cell-depleting therapies. Even though the direct evidence of TIX/CIL’s efficacy in lung transplant recipients is limited, the available data support the beneficial effects of this combination of mAbs in reducing infection, severe diseases, and adverse outcomes in this population. TIX/CIL acts, synergically, with a unique mechanism of action. SARS-CoV-2 enters human cells by the interaction of the spike glycoprotein with the angiotensin-converting enzyme 2 (ACE2) of the host cells. Targeting two different sites of the spike protein of SARS-CoV-2, the coadministration of TIX and CIL inhibits the attachment of the virus onto the cell surface preventing the infection [17].

TIX/CIL was authorized by the Food and Drug Administration (FDA) and the European Medicines Agency (EMA) for emergency use for the prevention of COVID-19 in adults and pediatric patients with risk factors for progression and expected low response to vaccination. In Italy, TIX/CIL was authorized for pre-exposure prophylaxis of SARS-CoV-2 infection in adults by the Italian Minister of Health in January 2022. However, its use worldwide has been limited by the extensive spread of the Omicron variants that have mutations in the spike protein that reduce susceptibility to TIX/CIL and increase transmissibility. In fact, in vitro data show decreased activity against omicron BA.4/5, BQ.1.1, and XBB.1.5 variants [18,19,20]. The FDA restricted the use of TIX/CIL to countries where the frequency of non-susceptible SARS-CoV-2 variants is less than or equal to 90%. However, recently, several authors have reported the clinical efficacy of TIX/CIL during the Omicron surge despite in vitro decreased activity. For some variants, including Omicron variants BA.1 and BA.1.1, a double dosage (300/300 mg intramuscular administration) was recommended.

The aim of this study is to evaluate the clinical efficacy and safety of TIX/CIL in LTRs during the circulation of the Omicron BA.2/BA.5 sub-lineages.

## 2. Materials and Methods

This is a single-center prospective cohort study conducted at The Mediterranean Institute for Transplantation and Highly Specialized Therapies (ISMETT) a solid organ transplant center located in Palermo, Italy. In June 2022, SARS-CoV-2 PrEP with intramuscular administration of TIX/CIL 150/150 mg was offered to all LTRs with active follow-up. For the purpose of this study, LTRs who received TIX/CIL were compared to LTRs who did not for different reasons. The reasons for not receiving TIX/CIL included refusal of the patient, logistic difficulties, or presence of medical conditions associated with adverse events such thromboembolic events, thrombocytopenia, or coagulation disorders. Patients were included if they were at least 18 years of age and were LTRs. The primary outcome of this study was to evaluate the incidence of breakthrough SARS-CoV-2 infections, defined as a newly positive polymerase chain reaction (RT-PCR) or antigen test after at least ten days from administration of TIX/CIL. Secondary outcomes included: days to viral clearance of SARS-CoV-2, severity of COVID-19 disease, death from COVID-19, and occurrence of adverse events.

The following data were entered in a dedicated electronic database: age, sex, comorbidities, including diabetes, cardiovascular diseases, chronic obstructive pulmonary disease, chronic liver diseases, solid malignancy, hematological malignancy, chronic kidney disease, HIV infection/AIDS, rheumatic diseases, age-adjusted Charlson comorbidity index [21], history of transplant rejection, SARS-CoV-2 vaccination, and number of doses received, SARS-CoV-2 serology prior to TIX/CIL when available, SARS-CoV-2 infection prior and after TIX/CIL, COVID-19 severity according to WHO criteria (asymptomatic if they tested positive for SARS-CoV-2 but without development of symptoms, mild in the case of COVID-19 symptoms such as fever, cough, sore throat, moderate in the case of respiratory failure requiring supplemental oxygen therapy, severe in the case of severe respiratory failure requiring mechanical ventilation), length of SARS-CoV-2 positivity, need for supplemental oxygen therapy, need for hospitalization, need for Intensive Care Unit (ICU) admission, and attributable and overall mortality. Regarding immunosuppression, at our institution, lung transplant recipients are treated with methylprednisolone and basiliximab as induction regimen, and tacrolimus, mycophenolate, and prednisone as maintenance regimen. Before receiving TIX/CIL, patients were tested for SARS-CoV-2 infection via antigen test of RT-PCR, as part of our hospital protocol before medical visits. A follow-up assessment was performed at months one and three after the TIX/CIL offer in order to evaluate rates of SARS-CoV-2 infection, hospitalization, mortality, and adverse effects.

Diagnosis of COVID-19 was performed on the basis of a positive RT-PCR or antigen test for SARS-CoV-2 performed on either nasopharyngeal throat swab or lower respiratory tract specimens.

Data are presented as median and interquartile range (IQR) for quantitative parameters and absolute numbers and percentages for categorical variables. Comparison between the group of patients who did and did not receive TIX/CIL was investigated by Mann–Whitney test and chi-squared or Fisher’s exact test. The association between TIX/CIL and SARS-CoV-2 infection was analyzed by fitting a multivariable logistic regression analysis, adjusted for TIX/CIL administration, prior SARS-CoV-2 infection, SARS-CoV-2 vaccination, age, years from transplant, and episodes of rejection. A *p*-value < 0.05 was considered as statistically significant. Statistical analyses were performed with STATA software (v14; StataCorp, College Station, TX, USA).

This study was approved by the IRCCS-ISMETT Institutional Research Review Board (IRRB/21/22) and was conducted according to the guidelines of the Declaration of Helsinki. All participants signed a written informed consent for the use of their anonymized data for research purposes.

## 3. Results

During the study period, 110 LTRs were enrolled in this study. A total of 79 (72%) received TIX/CIL and 31 (28%) did not.

Background demographics, including age, sex, Charlson Index, and history of rejection, were similar in both groups (Table 1). Almost 70% of the study population was male with a median age of 55 years and a median Charlson score of 2. Comorbidities were equally distributed among the two groups. In particular, cardiovascular diseases were present in 21.5% of the patients in the TIX/CIL group as compared to 12.9% in the control group (*p* = 0.3). Half of the patients of our cohort experienced at least one episode of rejection, with no difference among groups (57% vs. 42%, *p* = 0.16).

Regarding SARS-CoV-2-related variables, patients in both groups had similar rates of up to three doses of SARS-CoV-2 vaccination (97.5% vs. 90%, *p* = 0.11) and SARS-CoV-2 infection before TIX/CIL offer (46% vs. 26%, *p* = 0.06). Patients in the TIX/CIL group had a higher rate of positive SARS-CoV-2 serology before PrEP (49% vs. 23%, *p* = 0.03) (Table 1).

Overall, 14 (12%) SARS-CoV-2 infections were observed in the whole population. In particular, breakthrough SARS-CoV-2 infections occurred in 6% (5/79) of the patients in the TIX/CIL group compared to 29% (9/31) of patients in the control group (*p* < 0.001). The majority of the infections in both groups occurred within one month after the TIX/CIL offer and after a median of 7 years from transplant. The median time from TIX/CIL to SARS-CoV-2 infection was 18 days (IQR 11–42).

All patients had asymptomatic or mild infection, and no one was hospitalized or died of COVID-19. There was no difference in viral clearance between groups (15 vs. 17 days, *p* = 0.64) (Figure 1).

The adverse events related to TIX/CIL treatment occurred in only four (5%) patients and were all mild, including dry cough, palpitations, and hyperglycemia, and unrelated to the TIX/CIL administration.

The factors potentially associated with SARS-CoV-2 infection were studied by multivariable logistic regression analysis, adjusted for TIX/CIL administration, prior SARS-CoV-2 infection, SARS-CoV-2 vaccination, age, years from transplant, and episodes of rejection. At univariate regression analysis, TIX/CIL administration resulted in being the only factor associated with a significantly lower risk of infection (OR 0.16, 0.05–0.54 CI, *p* = 0.01). This was confirmed at multivariable analysis, after adjusting for confounders, which included age, years from transplant, history of rejection, prior SARS-CoV-2 infection, and vaccination (aOR 0.22, 00.6–0.78 CI, *p* = 0.02) (Table 2).

## 4. Discussion

In our cohort, we reported our favorable experience with TIX/CIL in lung transplant recipients. Treated patients had a significantly lower rate of SARS-CoV-2 infection and TIX/CIL PrEP resulted in being independently associated with a reduction in risk of infection. Despite the availability of vaccines, SARS-CoV-2 still has a significant impact on morbidity and mortality in LTRs [22,23], due to their inability to mount an adequate response to the vaccine that results in a lower quantity of antibodies produced with less neutralization activity and a faster decline in title over time [24,25,26]. It has been proven that lower antibody levels are associated with increased risk of hospitalization due to COVID-19. In the study of Malahe et al. in a cohort of immunocompromised hosts (solid organ and stem cell transplant recipients, patients on anti-CD20 therapy), antibody levels were measured after SARS-CoV-2 infection with Omicron variants and low levels of IgG resulted in being associated with a high risk of hospitalization [27]. Thus, there is a great need for alternative strategies to reduce the burden of SARS-CoV-2 infection.

The efficacy of the prophylactic use of this combination of mAbs in reducing COVID-19 and unfavorable complications has already been proven by several studies including high quality systematic reviews and meta-analyses [7,8], but fewer studies have focused specifically on solid organ transplant recipients [9,10]. In this study, we report our use of TIX/CIL for PrEP in LTRs during the Omicron wave. We found that LTRs who received TIX/CIL had a significantly lower incidence of SARS-CoV-2 breakthrough infection. Our findings are consistent with the recent work of Al Jurdi et al. [16] who reported a significantly lower infection rate among solid organ transplant recipients who received TIX/CIL (5% vs. 14%) during the BA.1 and BA.2 waves. It has to be acknowledged that in the study of Al Jurdi et al., the effect of TIX/CIL may have been overestimated due to the effect of the immortal time bias as reported by Riggs et al. in their letter to the Editor of the American Journal of Transplantation [28]. This bias is inherent with observational studies.

A positive effect of TIX/CIL PrEP during the Omicron wave has been reported by Al-Obaidi et al. in a cohort of 463 patients including 85 of SOT and hematopoietic stem cell transplant recipients [29]. Although a control group was not present, the rates of SARS-CoV-2 infection and related hospitalization were low [29]. Grillini et al. conducted a single center retrospective study that compared the incidence of infection in a group of 136 LTRs who received TIX/CIL and in 150 LTRs who did not. SARS-CoV-2 infection occurred in 8% of patients in the TIX/CIL group and in 34% in the untreated group with an 83% risk reduction (*p* < 0.001) [30]. Conversely, some authors have reported negative results with the use of TIX/CIL with a higher incidence of infection after TIX/CIL as compared to the incidence reported in the PROVENT trial. In the retrospective study of Morado et al., conducted in a cohort of 90 SOT recipients including 34 LTRs, 45 patients who received TIX/CIL were matched with 45 patients who did not receive PreEP, The administration of TIX/CIL was not effective in reducing rates of SARS-CoV-2 although a trend toward a lower number of infections in the TIX/CIL was observed (7% in the TIX/CIL and 18% in the control group, *p* = 0.2) [31]; moreover, they also reported lower numbers of hospitalization, and all-cause mortality in the case group. The authors’ hypothesis about the poor efficacy shown of TIX/CIL is that their study was conducted in a phase of the pandemic when some non-susceptible variants were circulating (BA.5) as compared to the PROVENT that was conducted when Alpha and Delta were the predominant variants. Another single center retrospective study, conducted in the U.S. by Sindu et al., compared 203 LTRs who received TIX/CIL with 343 who did not. A trend toward lower rate of infection (12% vs. 17%) and hospitalization (21% vs. 43%) were observed in the TIX/CIL group. However, after performing propensity-matched analysis in a subpopulation of 34 LTRs, risk of hospitalization, ICU admission, and mechanical ventilation was similar among the two groups. An explanation of these findings could be attributed, based on the authors’ conclusions, to the lower efficacy of TIX/CIL versus Omicron variants [32]. Despite these studies reporting negative outcomes, the study of Nguyen et al., that included one of the largest cohort of immunocompromised patients prophylactically treated with TIX/CIL 150 + 150 mg during the Omicron wave, described a positive effect. In fact, among 1112 patients, only 49 developed COVID-19 (4.4%) with an incidence rate lower than the one reported in the same region and in the same time period in the general population [6].

It is worth mentioning that TIX/CIL has also been studied for the treatment of COVID-19 in unvaccinated patients with SARS-CoV-2 infection, instead of prophylaxis. In the sponsored phase III randomized controlled trial named TACKLE, TIX/CIL at the dosage of 300 + 300 mg was administrated to 456 unvaccinated non-hospitalized patients with documented SARS-CoV-2 infection and compared with patients who received placebo. Progression to severe COVID-19 or death occurred in 4% of patients in the treatment group as compared to 9% in the placebo group with an absolute risk reduction of 4.5% (*p* < 0.001). These findings were confirmed in a recent systematic review and meta-analysis from Glhoom et al., in which the therapeutic use of TIX/CIL reduced the mortality rate compared to placebo [9]. However, since then, TIX/CIL has been used mainly for prevention possibly due to more effective therapeutic options.

The appropriate dosage of TIX/CIL as prophylaxis in LTRs is debated. In our study, patients received TIX/CIL at the standard 150/150 mg dosage as recommended at that time. However, due to the reduced activity of TIX/CIL against Omicron sublineages, the Food and Drug Administration (FDA) authorized the use of an increased dose (300/300 mg). There is limited evidence regarding the effectiveness of the increased dosage. A recent systematic review and meta-analysis from Glhoom et al. evaluated the efficacy of TIX/CIL at standard and double dose in immunocompromised hosts, mainly kidney transplant recipients [9]. TIX/CIL was effective both for the prevention and treatment of COVID-19, but without clear benefit of the increased TIX/CIL dose for all the outcomes assessed except for reducing the number of SARS-CoV-2 infections. In the already discussed study by Sindu et al., despite administration at the increased dosage, TIX/CIL did not reduce the incidence of SARS-CoV-2 infection [32].

In January 2023, the FDA limited the use of TIX/CIL since the overall circulation of in vitro non-susceptible variants was greater than 90% in the United States [33]. In vitro resistance of Omicron variants is due to spike codon substitutions that reduce the susceptibility to TIX/CIL, as recently described by Ordaya et al. [34]. As a consequence of the circulation of these variants, currently, TIX/CIL use for PrEP of COVID-19 is no longer recommended by the latest guidelines [35,36,37,38]. However, the clinical relevance of in vitro data is not always clear and has to be confirmed by in vivo studies. Indeed, clinical efficacy of TIX/CIL during Omicron surge was reported by several authors despite in vitro resistance. This is probably due to the high serum levels of monoclonal antibodies reached after TIX/CIL administration, up to 1000-fold greater, that exceed the concentration needed to achieve in vivo efficacy, as Solera et al. recently observed [39]. The authors performed a prospective study to evaluate the neutralization effects of TIX/CIL collecting the serum of 79 solid organ transplant recipients, including 20 LTRs; they evaluated the presence of neutralization 3 weeks after administration of TIX/CIL and found that almost all patients were able to neutralize BA 4/5 despite in vitro resistance. Interestingly, the authors concluded that in cases of low level of resistance, as for BA 4/5 but not for BQ.1.1, TIX/CIL may still have clinical efficacy. Jordan et al. recently published a post hoc analysis on the serological efficacy of PrEP in 911 SOTr who received TIX/CIL in periods when the Omicron BA and BQ subvariants were circulating [10]. Based on their findings, TIX/CIL was associated with a significant increase in the anti-spike IgG antibody which resulted in lower infection risk. However, only 10% of the population were lung transplant recipients.

Unfortunately, there are no other options for PrEP with mAbs and prevention of SARS-CoV-2 infection relies solely on general precautions, vaccination, and rapid seeking of medical attention in case of suggestive signs and symptoms. SARS-CoV-2 continues to evolve, and new variants are monitored to evaluate the potential impact on existing therapy and to develop new medical products against these variants. Currently, a new formulation of monoclonal antibody variants is under evaluation.

With regards to safety, TIX/CIL administration was well tolerated in our study and adverse effects were reported by only four patients. Of note, no cardiovascular adverse effects were observed while a worrisome number of serious cardiac effects (including myocardial infarction and arrhythmia) were noted in a post hoc analysis of the PROVENT Trial [5]. Overall, TIX/CIL was found to have, both in this and in previous studies, fewer and milder adverse effects than expected despite initial concerns [16,40]. The majority of the real world studies, including the already mentioned study by Morado et al., described mild or no adverse effects associated with TIX/CIL use [31].

We acknowledge that there are limitations to our study. This study is observational with the well-known selection biases; the control group was not matched, but all the most important potential confounders were included in the multivariate model. Similarly, this study can only make conclusions about the SARS-CoV-2 variants that were prevalent in Italy during the time period of this study (BA.2/BA.5) [41]. In our cohort, in patients with SARS-CoV-2 infection, causative variants were not studied; therefore, it is possible that some patients were also infected with variants other than BA.2/BA.5, not susceptible to TIX/CIL. Lastly, we only included lung transplant recipients in our cohort. However, the exclusive focus on LTRs can be also considered as a strength of this study because of the more homogenous population. In addition, LTRs are the most at-risk patients for complications related to SARS-CoV-2 infection.

Despite these limitations and the withdrawal of TIX/CIL from the market in many countries, our study adds real-life evidence of the efficacy and safety of this combination of mAbs for pre-exposure prophylaxis in vaccinated lung transplant recipients during the Omicron wave. The increasing population of immunocompromised hosts, including not only solid organ or stem cell transplant recipients but also patients in end-stage organ diseases, with hematologic or solid malignancies or other type of primary immunodeficiencies, remains at risk for COVID-19-related complications despite the evolution of the virus toward less virulent variants and advances in prevention and therapeutic strategies. Moreover, we highlight the importance of COVID-19 management strategies based on prevention rather than treatment of the infection, especially in immunocompromised hosts expected to respond poorly to vaccinations. Nevertheless, since SARS-CoV-2 continues to evolve, it cannot be excluded that new variants again become susceptible to TIX/CIL or that a new combination of neutralizing antibodies will be formulated allowing reconsideration of SARS-CoV-2 pre-exposure prophylaxis.

## Figures and Tables

**Figure 1 microorganisms-12-01436-f001:**
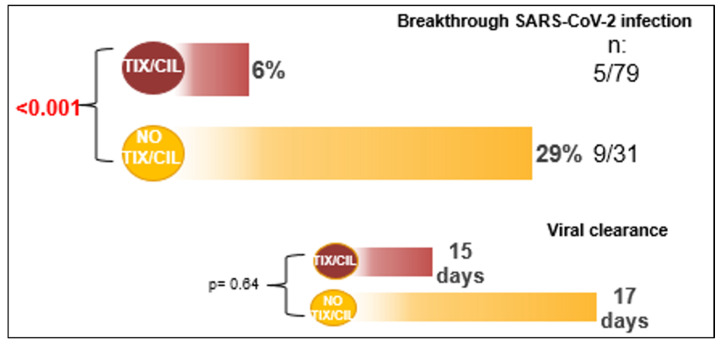
Outcomes of patients.

**Table 1 microorganisms-12-01436-t001:** Demographic and clinical characteristics of patients.

	Patients Treated with TIX/CIL 150 + 150 mg n = 79 (72%)	Patients Who Did Not Receive TIX/CIL n = 31 (28%)	*p* Value	Overall n = 110
Gender, Male, n (%)	52 (65.8%)	24 (77.4%)	0.24	76 (69.1%)
Age, years, median (IQR)	56 (45–64)	52 (39–61)	0.33	55.5 (42–64)
Age Unadjusted Charlson score, median (IQR)	2 (1–4)	2 (0–4)	0.28	2 (1–4)
Comorbidities, n (%)				
Diabetes	19 (24.1%)	8 (25.8%)	0.85	27 (24.5%)
Cardiovascular diseases	17 (21.5%)	4 (12.9%)	0.30	21 (19.1%)
COPD	30 (37.9%)	9 (29%)	0.38	39 (35.5%)
Chronic liver diseases	4 (5.1%)	0 (0%)	0.20	4 (3.6%)
Solid malignancy	3 (3.8%)	1 (3.2%)	0.86	4 (3.6%)
Hematological malignancy	2 (2.5%)	0 (0%)	0.37	2 (1.8%)
Chronic kidney disease	12 (15.2%)	6 (19.4%)	0.56	18 (16.4%)
HIV infection/AIDS	1 (1.3%)	0 (0%)	0.53	1 (0.9%)
Rheumatic Diseases	4 (5.1%)	2 (6.4%)	0.77	6 (5.5%)
Rejection, y	45 (56.9%)	13 (41.9%)	0.16	58 (52.7%)
Prior SARS-CoV-2 infection, y	36 (45.6%)	8 (25.8%)	0.06	44 (40%)
SARS-CoV-2 vaccination, y	77 (97.5%)	28 (90.3%)	0.11	105 (95.5%)
≥doses	65 (82.3%)	22 (70.9%)	0.21	87 (79.1%)
SARS-CoV-2 serology			0.03	
Negative	14 (17.7%)	10 (32.6%)		24 (21.8%)
Positive	39 (49.4%)	7 (22.6%)		46 (41.8%)
Unavailable	26 (32.9%)	14 (45.2%)		40 (36.4%)
SARS-CoV-2 Infection	5 (6.3%)	9 (29%)	<0.001	14 (12.7%)
Infection at 30-days	4/5 (80%)	5/9 (55.6%)		9/14 (64.3%)
Infection at 90-days	1/5 (20%)	4/9 (44.4%)		5/14 (35.7%)
Time from transplant to SARS-CoV-2 infection (years), median (IQR)	5.0 (1.3–9.8)	7.7 (1.4–10.7)	0.56	7.1 (1.2–10.8)
Time from TIX/CIL to SARS-CoV-2 infection (days), median (IQR)	18 (11–42)	NA		NA
COVID-19 Severity			0.30	
Asymptomatic	1/5 (20%)	4/9 (44.4%)		5/14 (35.7%)
Mild	4/5 (80%)	5/9 (55.6%)		9/14 (64.3%)
Moderate	0 (0%)	0 (0%)		0 (0%)
Severe	0 (0%)	0 (0%)		0 (0%)
Length of SARS-CoV-2 positivity (days), median (IQR)	15 (7–18)	17 (10–20)	0.64	16.5 (7–20)
Need for O2-therapy	0 (0%)	0 (0%)		0 (0%)
Hospitalization	0 (0%)	0 (0%)		0 (0%)
ICU admission	0 (0%)	0 (0%)		0 (0%)
Death due to COVID-19	0 (0%)	0 (0%)		0 (0%)
Non attributable mortality	1 (1.3%)	0 (0%)	0.23	1 (0.9%)

AIDS = acquired immune deficiency syndrome; COPD = chronic obstructive pulmonary disease; HIV = human immunodeficiency virus; ICU = intensive care unit; IQR = interquartile range; SARS-CoV-2 = severe acute respiratory syndrome coronavirus 2; TIX/CIL = tixagevimab/cilgavimab.

**Table 2 microorganisms-12-01436-t002:** Uni- and multi-variate regression analysis of factors associated with SARS-CoV-2 infection.

Factors Investigated for Association with SARS-CoV-2 Infection	Univariate Analysis				Multivariate Analysis			
	OR	95%CI		*p* Value	aOR *	95%CI		*p* Value
TIX/CIL	0.16	0.05	0.54	0.01	0.22	0.06	0.78	0.02
Age, per 10 years older	0.79	0.53	1.18	0.25	0.68	0.41	1.15	0.15
Years from transplantation	1.02	0.93	1.13	0.44	1.03	0.92	1.15	0.63
Prior SARS-CoV-2 infection, y	0.37	0.96	1.39	0.14	0.37	0.08	1.69	0.19
Rejection, y	0.45	0.14	1.44	0.18	0.41	0.11	1.52	0.18
SARS-CoV-2 vaccination, y	0.57	0.06	5.45	0.62	1.16	0.78	17.17	0.91

* Adjusted for all the factors showed in table. aOR = adjusted odds ratio; CI = confidence interval; OR = odds ratio; SARS-CoV-2 = severe acute respiratory syndrome coronavirus 2; TIX/CIL = tixagevimab/cilgavimab.

## Data Availability

The data that support the findings of this study are available from the corresponding author upon reasonable request.

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
