# Peer review of "Tixagevimab/Cilgavimab as SARS-CoV-2 Pre-Exposure Prophylaxis in Lung Transplant Recipients during the Omicron Wave: A Real-World Monocentric Experience"

_microorganisms, 2024, doi:10.3390/microorganisms12071436_

Round 1

Reviewer 1 Report

Comments and Suggestions for Authors

The paper presented for review analyze SARS-CoV-2 pre-exposure prophylaxis with monoclonal antibodies in lung transplant recipients.

The Title of the article is clearly and precisely formulated, comprehensible and adequately reflects the content and topic of the research.

In the Abstract, the Authors described the aims of the study, methodology, the most important results of the study and conclusions derived from the results of the studyThe range of keywords covers all the issues discussed in the article.

The Introduction is written systematically and provides insight into the complexity of the lung transplant recipients who are at increased risk of contracting severe COVID 19 due to their weak immune system and the fact that vaccination is not so much effective. Authors highlights alternative profilaxis with combination of neutralizing monoclonal antibodies (tixagevimab/cilgavimab) and present the conclusions from different studies which suggest its clinical efficacy during Omicron circulation. 
However, I find this section somewhat brief and perhaps a part regarding TIX/CIL mechanism of action could be added to the introduction section.

 The area of ​​research is defined by aim of the study to evaluate the clinical efficacy and safety of mAb prophilaxis in lung transplant recipients.  

Materials and methods used in the research are presented and described.

Specific point: In this section (Materials and methods) should be part regarding study groups and inclusion/exclusion criteria and not in the Results (line 111-112: " Reasons for not receiving TIX/CIL included refusal of the patient, logistic difficulties or presence of medical contraindications.")

Have you conducted real time RT-PCR or antigen test for SARS-CoV-2 infections on all participants in the study prior to receiving TIX/CIL?

Results of the Author's own research are presented in the article on 2 Tables and 1 Figure and are accompanied by clear textual interpretations.

Specific points:

Line 112: "Reasons for not receiving TIX/CIL included refusal of the patient, logistic difficulties or presence of medical contraindications."

Beside the comment that this should be part of Materials and Metods, can you specify what were medical contraindications? Do you think that the mentioned contraindications put LTRs as more vulnarable individuals in the control group that did not receive TIX/CIL than dose who received ? Could this factor affect the final results of the study?

Table 1. You have data about SARS-CoV-2 infection after 30 and 90 days. However in Material and Metods you mentioned only follow up after 3 months:

Line 96-97: " A follow-up assessment was performed at three months after  TIX/CIL offer in order to evaluate rates of SARS-CoV-2 infection and adverse effects."

You should be more detailed in Materials and Metods what and when was performed on both study groups so that the Tables could follow the stated methodology and results.

The discussion is written in accordance with the obtained results, and the Authors draw attention to many important issues of their own study in correlation with the results obtained by other authors. The Authors highlighted limitations to the study as well as strength. 

Specific point: 

Line: 193-194: " Similarly, this study can only make conclusions about the SARS-CoV-2 variants that were prevalent in Italy during the time period of the study (BA.2/BA.5)"

Did you confirm that all your SARS-CoV-2 positive patients had those variants? If not, could it be that some other variants were not susceptible to TIX/CIL?

In their conclusions, the Authors emphasized the importance of conducting further studies to identify key factors for prevention of infection in immunocompromised patients. 

The analysis of the topics raised by the Authors has been presented in a clear and coherent manner. All items of scientific literature (26) are up-to-date and can be found in the text of the article. The language of the work is understandable and easy to read.

I rate this study very well in terms of content and recommend it for publication after additions indicated in the review.

Author Response

Point-by-point response Letter

-----------------

Dear Colleague,

We are now submitting a revised version of the manuscript entitled “Tixagevimab/cilgavimab as SARS-CoV-2 pre-exposure prophylaxis in lung transplant recipients during the Omicron wave: a real-world monocentric experience”, addressing all the queries raised.

------

R1: The paper presented for review analyze SARS-CoV-2 pre-exposure prophylaxis with monoclonal antibodies in lung transplant recipients. The Title of the article is clearly and precisely formulated, comprehensible and adequately reflects the content and topic of the research. In the Abstract, the Authors described the aims of the study, methodology, the most important results of the study and conclusions derived from the results of the study. The range of keywords covers all the issues discussed in the article.

Author response: We would like to thank the reviewer for his work.

The Introduction is written systematically and provides insight into the complexity of the lung transplant recipients who are at increased risk of contracting severe COVID 19 due to their weak immune system and the fact that vaccination is not so much effective. Authors highlights alternative prophylaxis with combination of neutralizing monoclonal antibodies (tixagevimab/cilgavimab) and present the conclusions from different studies which suggest its clinical efficacy during Omicron circulation. 
However, I find this section somewhat brief and perhaps a part regarding TIX/CIL mechanism of action could be added to the introduction section.

AR: We agree that the introduction was too brief. In the revised version of the manuscript, a more comprehensive introduction was added including TIX/CIL mechanism of action as suggested (P 2, L 86-90).

The area of ​​research is defined by aim of the study to evaluate the clinical efficacy and safety of mAb prophylaxis in lung transplant recipients. Materials and methods used in the research are presented and described. Specific point: In this section (Materials and methods) should be part regarding study groups and inclusion/exclusion criteria and not in the Results (line 111-112: " Reasons for not receiving TIX/CIL included refusal of the patient, logistic difficulties or presence of medical contraindications.")

AR: Thanks for your suggestion, we have modified the manuscript including reasons for not receiving TIX/CIL in the material and methods section as follow: “Reasons for not receiving TIX/CIL included refusal of the patient, logistic difficulties or presence of medical conditions associated with adverse events such thromboembolic events, thrombocytopenia or coagulation disorders”.

Have you conducted real time RT-PCR or antigen test for SARS-CoV-2 infections on all participants in the study prior to receiving TIX/CIL?

AR: Thank you for your comment on this important point that we have included in our material and methods section: “Before receiving TIX/CIL, patients were tested for SARS-CoV-2 infection by antigen test of RT-PCR, as part of our hospital protocol before medical visits.”

Results of the Author's own research are presented in the article on 2 Tables and 1 Figure and are accompanied by clear textual interpretations. Specific points:

Line 112: "Reasons for not receiving TIX/CIL included refusal of the patient, logistic difficulties or presence of medical contraindications." Beside the comment that this should be part of Materials and Methods, can you specify what were medical contraindications? Do you think that the mentioned contraindications put LTRs as more vulnarable individuals in the control group that did not receive TIX/CIL than dose who received? Could this factor affect the final results of the study?

AR: Thank you for your comment. As specified above, medical contraindications are now detailed in the manuscript. Regarding the second point, being an uncontrolled observational study, there is a well-known risk of selection bias for which some conditions could be more frequent in a group as compared to the other and this may affect the final results. However, in our study we have tried to control this bias performing logistic regression adjusted for confounders. In our cohort, comorbidities were equally distributed among the two groups and were not independently associated with SARS-CoV-2 (table 1 and 2). Therefore, we believe that these factors have not influenced our results.

Table 1. You have data about SARS-CoV-2 infection after 30 and 90 days. However in Material and Methods you mentioned only follow up after 3 months: Line 96-97: " A follow-up assessment was performed at three months after TIX/CIL offer in order to evaluate rates of SARS-CoV-2 infection and adverse effects." You should be more detailed in Materials and Methods what and when was performed on both study groups so that the Tables could follow the stated methodology and results.

AR: We have modified the manuscript according to your suggestion.

The discussion is written in accordance with the obtained results, and the Authors draw attention to many important issues of their own study in correlation with the results obtained by other authors. The Authors highlighted limitations to the study as well as strength. Specific point: 

Line: 193-194: "Similarly, this study can only make conclusions about the SARS-CoV-2 variants that were prevalent in Italy during the time period of the study (BA.2/BA.5)". Did you confirm that all your SARS-CoV-2 positive patients had those variants? If not, could it be that some other variants were not susceptible to TIX/CIL?

AR: Once again thanks for your comment. In our cohort, in patients with SARS-CoV-2 infection, causative variants were not studied; therefore, we agree with the reviewer that some patients may have been infected also with variants, other than BA.2/BA.5, not susceptible to TIX/CIL. This is now included in our discussion (P9, L322-325).

 In their conclusions, the Authors emphasized the importance of conducting further studies to identify key factors for prevention of infection in immunocompromised patients. The analysis of the topics raised by the Authors has been presented in a clear and coherent manner. All items of scientific literature (26) are up-to-date and can be found in the text of the article. The language of the work is understandable and easy to read. I rate this study very well in terms of content and recommend it for publication after additions indicated in the review.

AR: we would like to thank the reviewer for having reviewed our article and for the suggestions provided that we are sure will improve the overall quality of the manuscript.

We hope that we have properly addressed the Reviewer’s queries and that the novel data presented may be considered of interest to the readers of Microorganisms.

With our warmest regards

Andrea Cona and Alessandra Mularoni

Reviewer 2 Report

Comments and Suggestions for Authors

Dear Authors,

I carefully reviewed your manuscript, which describes a small clinical trial evaluating the Tixagevimab/Cilgavimab (TIX/CIL) mAB combination as a pre-exposure prophylactic for SARS-CoV-2 infection in lung transplant recipients (LTR). The study provides important statistical outcomes from a two-arm study involving 110 LTRs, with 79 recipients receiving TIX/CIL and 31 not receiving it. You provided very important comprehensive description of the clinical characteristics of the patients in the lengthy table.

However, I have few serious concerns about  readiness of this manuscript for scientific publication. Firstly, there is serious lack of a comprehensive review of the available information and a lack of discussion regarding the issues associated with TIX/CIL treatments and the differences in outcomes. We encourage you to incorporate a thorough review of published results, as there are several relevant publications that have not been mentioned, such as the study by Morado et al. in 2023 ; study by D. Sindu et al. in 2023; Al-Obaidi et al. published in the American Journal of Medicine in 2023; by A. Grillini et al in the Journal of Heart and Lung Transplantation in 2022

For instance in the discussion section,  line 155-158 you are comparing current results with results published by Al Jurdi et al, but you fail to mention the publication by D. Sindu et al. titled "Pre-exposure Prophylaxis with Tixagevimab-Cilgavimab did not Reduce Severity of COVID-19 in Lung Transplant Recipients with Breakthrough Infection" in the Journal of Transplantation direct, March 2023. And discuss the reason why that material demonstrates the absence of clinical effects in larger studies involving a population of 546 LTRs.

We suggest that you address the mention of "medical contradictions" associated with the use of TIX/CIL in lines 111-112. It would be important to describe these contradictions in either the introduction or discussion section.

Additionally, in addition to lines 172-174 , it would be beneficial to discuss the  tests and mechanisms by which TIX/CIL can provide effective protection in patients  from the virus that is showing resistance to these antibodies in vitro tests. This manuscript lack of any discussion of the scientific fundings with regards to the effectiveness of TIX/CIL as neutralizing antibody in vitro and in vivo. We recommend reviewing some published materials  to support this discussion.

While you have provided a good table of clinical characterization for the LTR group, you have not discussed any outcome results in associations with the described risk factors. Lack of this discussion making table is not as useful as it can be for further studies.

Lastly, I noted a few small issues: There is inconsistency in the reference index numbers in the text:  For example, line 72 is giving number [14-16], while line 74 is using number [10-12]. Additionally, the number [13. - Salvini M. et al.] is missing in the text.

Also, it would be helpful if headers could be added to the two columns in Table 2 for better clarity.

I hope that these suggestions and revision will improve the manuscript and contribute to its suitability for scientific publication. Please do not hesitate to reach out if you have any questions or need further clarification.

Author Response

Point-by-point response Letter

-----------------

Dear Colleague,

We are now submitting a revised version of the manuscript entitled “Tixagevimab/cilgavimab as SARS-CoV-2 pre-exposure prophylaxis in lung transplant recipients during the Omicron wave: a real-world monocentric experience”, addressing all the queries raised.

------

I carefully reviewed your manuscript, which describes a small clinical trial evaluating the Tixagevimab/Cilgavimab (TIX/CIL) mAB combination as a pre-exposure prophylactic for SARS-CoV-2 infection in lung transplant recipients (LTR). The study provides important statistical outcomes from a two-arm study involving 110 LTRs, with 79 recipients receiving TIX/CIL and 31 not receiving it. You provided very important comprehensive description of the clinical characteristics of the patients in the lengthy table.

AR: we would like to thank the reviewer for having reviewed our work and for the comments provided.

However, I have few serious concerns about readiness of this manuscript for scientific publication. Firstly, there is serious lack of a comprehensive review of the available information and a lack of discussion regarding the issues associated with TIX/CIL treatments and the differences in outcomes. We encourage you to incorporate a thorough review of published results, as there are several relevant publications that have not been mentioned, such as the study by Morado et al. in 2023 ; study by D. Sindu et al. in 2023; Al-Obaidi et al. published in the American Journal of Medicine in 2023; by A. Grillini et al in the Journal of Heart and Lung Transplantation in 2022

AR: thanks for the suggestions. We have reviewed each of the articles suggested among others works. We decided to discuss these articles in our manuscript in order to provide a more extensive review including articles with differences in outcome following TIX/CIL. 

For instance in the discussion section, line 155-158 you are comparing current results with results published by Al Jurdi et al, but you fail to mention the publication by D. Sindu et al. titled "Pre-exposure Prophylaxis with Tixagevimab-Cilgavimab did not Reduce Severity of COVID-19 in Lung Transplant Recipients with Breakthrough Infection" in the Journal of Transplantation direct, March 2023. And discuss the reason why that material demonstrates the absence of clinical effects in larger studies involving a population of 546 LTRs.

AR: We have now included and discuss the suggested study.

We suggest that you address the mention of "medical contradictions" associated with the use of TIX/CIL in lines 111-112. It would be important to describe these contradictions in either the introduction or discussion section.

AR: Thanks for your suggestion, we have modified the manuscript including reasons for not receiving TIX/CIL in the material and methods section as follow: “Reasons for not receiving TIX/CIL included refusal of the patient, logistic difficulties or presence of medical conditions associated with adverse events such thromboembolic events, thrombocytopenia or coagulation disorders”.

Additionally, in addition to lines 172-174 , it would be beneficial to discuss the  tests and mechanisms by which TIX/CIL can provide effective protection in patients  from the virus that is showing resistance to these antibodies in vitro tests. This manuscript lack of any discussion of the scientific fundings with regards to the effectiveness of TIX/CIL as neutralizing antibody in vitro and in vivo. We recommend reviewing some published materials to support this discussion.

AR: we have specified the mechanisms of in vitro resistance citing, among others, the work from Ordaya et al. More importantly we have included some theories for the in vivo efficacy despite in vitro resistance as the one described in the work from Solera et al. Thanks for your comment.

While you have provided a good table of clinical characterization for the LTR group, you have not discussed any outcome results in associations with the described risk factors. Lack of this discussion making table is not as useful as it can be for further studies.

AR: In the present work we have included exclusively patients that underwent lung transplantation, both in the treatment that in the control group. Therefore, we can draw conclusions only about this population. However, with our logistic regression model, we were able to confirm that TIX/CIL was the only factor independently associated with a reduction of risk of SARS-CoV-2 while other confounders had no positive or negative association. Other conditions at risk, such as comorbidities, were equally distributed in the two group. Data on the efficacy of TIX/CIL in other group of immunocompromised patients can be extrapolated from the literature. This was specified in the introduction and in the discussion.

Lastly, I noted a few small issues: There is inconsistency in the reference index numbers in the text:  For example, line 72 is giving number [14-16], while line 74 is using number [10-12]. Additionally, the number [13. - Salvini M. et al.] is missing in the text.

AR: Thanks. Reference list and index numbers have been corrected.

Also, it would be helpful if headers could be added to the two columns in Table 2 for better clarity.

AR: We have modified the table according to your suggestion.

We hope that we have properly addressed the Reviewer’s queries and that the novel data presented may be considered of interest to the readers of Microorganisms.

With our warmest regards

Andrea Cona and Alessandra Mularoni
